# BiVi-GAN: Bivariate Vibration GAN

**DOI:** 10.3390/s24061765

**Published:** 2024-03-08

**Authors:** HoeJun Jeong, SeongYeon Jeung, HyunJun Lee, JangWoo Kwon

**Affiliations:** 1Department of Electric Computer Engineering, Inha University, Incheon 22212, Republic of Korea; lilmae@inha.edu (H.J.); ran22314@inha.edu (S.J.); 2Technology Research Center, RMS Technology Co., Ltd., Cheonan 31217, Republic of Korea; hjlee@rmstech.co.kr; 3Department of Computer Engineering, Inha University, Incheon 22212, Republic of Korea

**Keywords:** vibration, rotary machine, deep learning, PINN, GAN

## Abstract

In the domain of prognosis and health management (PHM) for rotating machinery, the criticality of ensuring equipment reliability cannot be overstated. With developments in artificial intelligence (AI) and deep learning, there have been numerous attempts to use those methodologies in PHM. However, there are challenges to applying them in practice because they require huge amounts of data. This study explores a novel approach to augment vibration data—a primary component in traditional PHM methodologies—using a specialized generative model. Recognizing the limitations of deep learning models, which often fail to capture the intrinsic physical characteristics vital for vibration analysis, we introduce the bivariate vibration generative adversarial networks (BiVi-GAN) model. BiVi-GAN incorporates elements of a physics-informed neural network (PINN), emphasizing the specific vibration characteristics of rotating machinery. We integrate two types of physical information into our model: order analysis and cross-wavelet transform, which are crucial for dissecting the vibration characteristics of such machinery. Experimental findings show the effectiveness of our proposed model. With the incorporation of physics information (PI) input and PI loss, the BiVi-GAN showed a 70% performance improvement in terms of JS divergence compared with the baseline biwavelet-GAN model. This study maintains the potential and efficacy of complementary domain-specific insights with data-driven AI models for more robust and accurate outcomes in PHM.

## 1. Introduction

In industrial areas, numerous dynamic systems operate in an integrated manner, among which rotating equipment constitutes one of the most utilized types of machinery. Rotating equipment is predominantly employed in various mechanical conversion devices such as automobile engines, pumps, wind turbines, power generation systems, and gas turbines [1,2]. The failure of such equipment can potentially lead to malfunctions in the associated machinery and a loss of functionality in the overall system. Extensive research has been conducted on the diagnosis and prediction of the reliability and safety of rotating equipment [3,4,5,6]. Those approaches have evolved into the field of prognosis and health management (PHM). Particularly, research on artificial intelligence (AI)-based PHM has been actively conducted, owing to advancements in computer technology [7,8,9,10]. In the realm of AI-based PHM research, in contrast with traditional PHM approaches, which are structured around physical laws, various forms of data are acquired from the equipment. This data serves as the foundation for analyzing the equipment’s health and remaining useful life (RUL). With advancements in deep learning models, analytical methodologies have been proposed. Several studies are based on models that enable generalized analysis without the requirement for expert knowledge [11,12]. Such research endeavors utilize diverse data sources, including thermal images, high-speed camera images, vibration data, and noise data, to train AI models. Among these, vibration data have traditionally been a critical element in the analysis of rotating equipment within conventional PHM methodologies. Owing to its rich information related to the operation and condition of the equipment, vibration data are also extensively employed in deep learning studies where features contained in the data are of significant importance. However, a salient challenge with these data-driven methodologies is the conflict between the massive volume of data generally required to train AI models and the limited amount of vibration data that can realistically be collected in industrial settings.

As the prevalence of data-hungry deep learning models continues to rise, data augmentation techniques have increasingly become a significant issue in the field of AI. Numerous studies have demonstrated that training AI models on augmented data can help them outperform those trained on the limited amount of original data that can be collected in practice [13]. Vibration data have temporal correlations and can thus be classified as time series data. Studies on methods for augmenting time series data have been vigorously conducted, reflecting the growing recognition of its importance [14]. In basic approaches, various techniques such as cropping, flipping, and jittering are employed in the time series, frequency, and time–frequency domains to artificially augment the data [15]. These methodologies do not focus on understanding the inherent characteristics of the data; rather, they augment the data based on specific rules and probabilities. Recently, with the development of data analytics and learning models, techniques for analyzing the complex rules and information inherent in data have emerged [16,17]. These include decomposition methods, statistical generative models, and learning models. Particularly, deep learning-based generative models such as generative adversarial networks (GANs), variational autoencoders (VAEs), and diffusion models have high applicability across various domains. This is because they can extract the nonlinear features inherent in the data in an end-to-end manner without the requirement for specialized knowledge.

However, the black-box nature of deep learning models poses a challenge, as these generative models may not adequately capture the key physical characteristics relevant to vibration analysis [18,19]. Consequently, there are frequent instances where the generated data fail to reflect the vibration characteristics required for analyzing rotating equipment. To address these challenges raised from the perspective of field application, physics-informed neural networks (PINNs) have emerged as a solution aimed at overcoming the limitations of deep learning models, which are highly dependent on the data themselves [20]. PINNs employ a structure that guides the neural network to learn physical information and characteristics that are difficult for AI to extract by itself. This enables the leveraging of both the advantages of deep learning, which has an excellent ability to extract various features based on data, and the robust, dimensionless characteristics that can be represented by physical equations. In this study, we aim to propose a vibration data augmentation model that incorporates the structure of a PINN. Specifically, the model is designed with a structure specialized for the vibration characteristics of rotating equipment, thus enabling efficient consideration of the relevant physical features. The model proposed in this study has the objective of mapping two types of physical information onto the existing structure of a bidirectional GAN (Bidirectional GAN) [21]. One of the types of physical information aims to incorporate order analysis, a key analytical focus in rotating machinery, which analyzes harmonic frequency components based on the equipment’s rotational speed. The other type of physical information aims to utilize cross-wavelet transform, a methodology that analyzes the time–frequency axis correlation between two sets of vibration data.

## 2. Related Work

### 2.1. Time Series Data Generation Using Generative Models

Generally, research on methods for generating data to train AI models, particularly deep learning models, has been consistently conducted. Recently, there has been a proliferation of studies focused on utilizing deep learning generative models for data creation [22,23]. The most commonly used are VAEs and GANs. Both VAEs and GANs are similar in terms of learning the data distribution based on maximum likelihood estimation. However, VAEs belong to the explicit density model category, approximating the prior distribution of the model to estimate the data distribution. In contrast, GANs are part of the implicit density model category, which, instead of explicitly defining the model, iteratively performs sampling to converge to a specific probability distribution. VAEs have the advantage of being able to generate data similar to a given *x* by directly calculating the probability of *x* in the training set, as shown in Figure 1.

However, VAEs have the drawback of producing “blurred” generated outputs owing to the injection of noise and the imperfection of the variational bounds. Consequently, much of the recent generative research has been based on GAN models. GANs are composed of two neural networks, commonly referred to as the generator and discriminator, as shown in Figure 2.

The generator aims to produce data that are difficult to distinguish from the real data, while the discriminator’s role is to differentiate between genuine and generated data [24,25]. Although GANs have a more challenging training process compared with VAEs, they are capable of generating sharper outputs and can cover a broader range of distributions beyond the specific distributions exhibited by the given input data. These advantages have led to their widespread use in recent research. The model proposed in this study also aims to generate vibration data gathered from rotating equipment, leveraging the adversarial training approach inherent to GANs. In this study, among various GAN models, we utilize the Bidirectional GAN with a conditional GAN structure as the base model. Conditional GANs aim to overcome the limitation of the original GAN, which cannot learn different distributions for multiple labels when only sampling *z* from the latent space as input. To address this, conditional GANs incorporate class information along with *z* as additional input, enabling the GAN model to learn different distributions for each class. Conditional GANs have evolved in various directions, from the simple act of adding a condition *y* to using the image itself as a condition, as in the case of pix2pix. In this study, the condition is set to include the presence or absence of defects, as well as the type of defects in the rotating equipment. This allows the model to learn the distribution of data that occur under various conditions.

The Bidirectional GAN not only learns to generate *x* from *z* but also incorporates an additional encoder structure that learns the inverse mapping operation, projecting *x* back into the latent space as shown in Figure 3 [21].

The discriminator is trained using *x* and *z* as inputs, undergoing a process to align $p_{G}$ and $p_{E}$. The authors claim that this structure can overcome the saddle point problem, which is a common issue in GANs. Subsequent research has indeed used the BiGAN structure to synthesize more refined and higher-quality images. In this study, the generator is required to simultaneously produce both the vibration signal itself and the continuous wavelet transform (CWT) image. Therefore, a more stable training method is required. We have configured the model to resolve the saddle point problem and reach a global optimum by employing the inverse mapping technique proposed in BiGAN.

### 2.2. Frequency Feature Extraction Using Wavelet

The Fourier transform operates under the assumption that any given signal can be represented as a sum of sine waves with varying frequencies and amplitudes. It is a method used to decompose a given signal into its constituent frequency components and intensities, thereby enabling data analysis in the frequency domain. Inherently designed for frequency component analysis, the Fourier transform inevitably suffers from the loss of information along the time axis [26]. To overcome this limitation, the short-time Fourier transform (STFT) was proposed, as shown in Figure 4 [27]. STFT involves dividing a signal that changes over time into short time windows and applying the Fourier transform to each window. This allows for the extraction of frequency components present within each time segment, which are then accumulated. The results of STFT can be represented as an image with a time and frequency axis, making it suitable for signal representation. Consequently, numerous machine and deep learning models have been applied to analyze given vibration signals through STFT analysis.

The wavelet transform overcome the limitation of STFT, which suffers from a trade-off between time resolution and frequency resolution owing to its reliance on sine wave-based frequency decomposition [28]. A wavelet transform involves defining a waveform function with a limited temporal extent and using it to analyze signals as shown in Figure 5.

This allows for an improved time–frequency resolution trade-off. Consequently, the wavelet transform offers the advantage of enhancing time resolution for high-frequency components and frequency resolution for low-frequency components. This enables a more balanced resolution trade-off based on the nature of the signal’s frequency components. However, the process of shifting the waveform function and calculating the similarity for the given input signal in a wavelet transform, while scaling the waveform function, results in a significantly higher computational load compared with STFT. Consequently, the wavelet transform is limited in real-world industrial applications owing to issues related to real-time processing and memory constraints.

Certainly, research utilizing similar frequency analysis methods has been conducted using multivariate time series data as well [29,30]. However, in most studies, each signal is often treated as an independent entity, applying separate transformations to each signal and subsequently concatenating them on a channel-by-channel basis [31,32]. Alternatively, these transformations are applied to multiple received inputs and utilized accordingly. In this study, in contrast with analyses that conclude with a frequency analysis of individual signals, the aim is to apply CWT to two signals after performing frequency analysis for both. This is to subsequently analyze the correlation between the two signals within the frequency domain. CWT involves applying the same mother wavelet signal and analysis technique to both signals. Following the transformation, the two values are multiplied to generate the cross-wavelet spectrum. This spectrum showcases the correlation and phase difference between the two signals in the time–frequency domain. The input signals utilized in this study are vibration data collected from the same vibration source with a phase difference of 90°. Leveraging the advantages of CWT, which effectively represents the frequency-based correlation and its strength, the study aims to analyze the bivariate vibration data and interpret the interactions between them in the frequency domain.

### 2.3. Learning Physical Characteristics Using PINN

Certainly, AI models, particularly methodologies such as machine learning (ML) and deep learning (DL), profoundly rely on data-driven approaches. Consequently, the dependence on the additional information inherent in the actual data diminishes. However, in domains where the underlying physical system is well-understood, leveraging predefined rules as prior knowledge before training on gathered data can lead to achieving strong performance with a limited dataset. This approach can also prevent issues arising from the application of black-box models to hidden data, which is a concern owing to the nature of such models. Recently, research aiming to combine the advantages of data-driven and physics-based approaches has gained momentum. This approach is referred to as physics-informed machine learning (PIML) or PINN. PINN is structured into two main branches, as shown in Figure 6.

One branch involves enforcing known physical information onto the AI model [33,34,35]. This can be achieved through methodologies such as incorporating the upper and lower bounds of physics information into the neural network as regularization or utilizing loss functions that leverage the physics information. The other branch involves the neural network itself mapping physical information. This is achieved through a “physics-informed (PI) architecture”, where specific nodes or layers are designed to receive or learn the physical information [36,37]. In this study, the proposed approach involves incorporating a PI input that considers frequency characteristics into the Bidirectional GAN-based generative model. Additionally, the model generates cross-wavelet transform images based on learned features during data generation. The differences between these images are used as a loss function, referred to as the PI loss, to leverage PI.

### 2.4. Bearing Failure Detection via Deep Learning Model

Bearings are one of the critical components of rotating machinery, and the failure of bearings is so significant that it can lead to the shutdown of the entirety of the equipment. Accordingly, numerous studies have been conducted with approaches aiming to perform bearing fault detection. Early research relied on features extracted by experts, utilizing methods like Artificial Neural Networks (ANN), k-Nearest Neighbor, and Support Vector Machines (SVM) for classification and anomaly detection. The features were mainly collected based on frequency domain, including Short-Time Fourier Transform (STFT), Fast Fourier Transform (FFT), Hilbert Transform (HT), and Wavelet Transform (WT) [38,39]. However, these methodologies have limitations due to the necessity of preceding feature engineering, due to heavily relying on expert knowledge in selecting feature extraction techniques, and in determining the structure of models based on these to perform tasks.

With the development of deep learning, many studies have been conducted to utilize the capability of deep learning to extract features directly from data. In the field of bearing fault detection, research utilizing deep learning has been actively conducted [40,41,42]. Tand et al. studied a method of visualizing data for the analysis of low-speed rotating equipment [43]. Particularly, there have been many studies utilizing inputs from multiple sensor data simultaneously, including methods diagnosing bearing status using multiple sensors and a small number of filters [44], and methods mixing each sensor channel for robust learning [45].

In this study, we utilize data from two sensors attached to the bearing. We model the 90-degree phase difference between the two sensors as a feature. Additionally, we propose BiVi-GAN that synthesizes data using the frequency and temporal characteristics obtained, thereby preserving correlations that cannot be considered in single-sensor data only.

## 3. Methodology

In this study, we propose a new GAN model, the BiVi-GAN, which employs a Bidirectional GAN architecture combined with PI loss and PI input. The BiVi-GAN, as shown in Figure 7, comprises three sub-networks: the generator, encoder, and discriminator.

The generator accepts random noise *z* following a Gaussian distribution and a 1x2x feature $x_{phy}$ extracted from real vibration data *x*, producing synthetic vibration data *x*′ and synthetic cross-wavelet transform images cwt′. The encoder considers the real data *x* and a 1x2x feature $z_{phy}$ extracted from random noise *z* as inputs and generates an estimated value *z*′ for *z*. The discriminator performs the task of classifying between real and fake by accepting the following as inputs: random noise *z*, the set of data generated by the generator from *z* as {z,x}′, real data *x*, and random noise *z*′ generated by the encoder based on *x*, represented as {z′, x}.

The proposed model utilizes two major types of physical information related to vibration. One is the order frequency extracted from the vibration signal *x*. Order frequency is a feature commonly used in order analysis, a diagnostic tool extensively employed in the field of mechanical engineering that signifies frequency values that are multiples of the equipment’s rotational speed. Numerous studies in the field of mechanical engineering have demonstrated that order analysis not only distinguishes between the normal and abnormal states of rotating equipment but also can diagnose the cause and severity of anomalies. Hence, it’s evident that the order frequency contains substantial physical information regarding defects in vibration data. In this study, we extract the 1st and 2nd order frequency and directly input them into the generator as the physical information for the given vibration data *x*. This can be compared to the method of incorporating physical prior knowledge, a technique frequently employed in PINN series of models. The generator is tasked with learning about frequency characteristics of *x* that are not described by this prior knowledge, as well as information regarding noises originating both externally and internally from the system.

The other piece of physical information is the cross-wavelet transform data extracted from the vibration signal. This study analyzes the condition of equipment based on data simultaneously collected from two sensors, each with a phase angle of 90°. The phase difference in sensors collecting data from rotating equipment allows for capturing distinct vibration characteristics owing to the rotational trajectory made by the equipment’s axis. The method of diagnosing abnormalities in the axis and bearings of rotating equipment based on a 90-degree phase difference has been substantiated through numerous prior studies. In several studies based on rotating equipment, it was common to construct and analyze an orbit, which represents the trajectory of the rotating axis, based on the vibration information acquired from two orthogonal sensors. However, in this study, we use wavelet analysis, which offers richer information than the orbit and simultaneously represents both the frequency and time domains. Among various wavelet-based analytical methodologies, we employ CWT to visualize the frequency response correlation between the two vibration signals. This visualized data are then utilized as the physical information extracted from the given pair of vibration signals. In this study, we employ a strategy where the generator, while producing the vibration signal, concurrently generates a CWT image. Through this strategy, the generator is able to simultaneously learn information in the time domain from direct analysis of the vibration signals, and information in the time–frequency domain by comparing the CWT images. Essentially, in the proposed model, the order frequency information is used as the PI input for both the generator and encoder. To compute the PI loss, we utilize the CWT image to produce the CWT loss, which in turn is used to train both the generator and encoder.

### 3.1. Physics Guided Generator & Encoder

Typically, a GAN considers the distribution of *z* as input, with the objective of having the generator learn to produce outputs that closely resemble the desired distribution of *x*. The goal is for the discriminator to be unable to distinguish between these generated outputs and the real data. In this study, instead of learning a direct function from *z* to *x*′, we first extract prominent features from *x* through order analysis. Using these frequency characteristics, we generate a baseline for *x*′ termed $x_{base}$. This $x_{base}$ serves as an intermediate result, constructed solely based on the extracted features. While it may not be entirely identical to *x*, it encapsulates prior physical knowledge. Therefore, the BiVi-GAN’s generator learns the remaining frequency components and noise that $x_{base}$ fails to represent. That is, major features that can be clarified physically are utilized as prior knowledge, while more elusive phenomena and noises are learned using the data. The encoder follows the structure of BiGAN and is designed to perform a task that is entirely opposite to that of the generator. The encoder considers the real signal *x* as input and synthesizes *z*′. In this process, similarly to the generator, it also performs the task of extracting the primary frequency components from *z*.

While the primary goal of this study is to synthesize vibration data themselves, as an auxiliary task, it also undertakes the generation of cross-wavelet transform images. One reason for this is that during the learning process of the cross-wavelet transform, features are extracted that consider the correlation between the two signals. Consequently, when generating vibration signals, vibrations that consider these features are produced. Each signal is recognized not as an independent entity but as data with correlations. This enables the generation of *x*′ to produce more realistic data.

### 3.2. Theoretical Convergence

#### 3.2.1. BiGAN with Auxiliary Loss

In this section, we will write generator, encoder, and discriminator as G,E, and *D*, respectively, and $L_{1}$ indicates L-1 Loss. The mini-max problem of the conventional BiGAN structure is expressed in Equation (Equation 1) as follows [21]:(1)$$\begin{array}{l} \min_{G,E}\max_{D}V\left(D,E,G\right)=\\ \min_{G,E}\max_{D}E_{\left(x,z\right)∼p_{EX}}\left[\log\left(D\left(x,E\left(x\right)\right)\right)\right]\\ \;+E_{\left(x,z\right)∼p_{GZ}}\left[\log\left(1-D\left(G\left(z\right),z\right)\right)\right] \end{array}$$
The probability distribution of (x,z) follows the distribution, as expressed in Equation (Equation 2).
(2)$$\begin{array}{c} P_{EX}\left(x,z\right)=P_{E}\left(z|x\right)P_{x}\left(x\right)\\ P_{GZ}\left(x,z\right)=P_{G}\left(x|z\right)P_{z}\left(z\right) \end{array}$$

#### 3.2.2. Optimal Discriminator

For fixed *G* and *E*, we can identify optimal discriminator *D*. To identify optimal *D*, we define $D^{*}$ as an optimal *D*.

**Theorem 1.** 

$D^{*}=\frac{p_{G}\left(x\right)\cdot p_{EX}\left(x,z\right)}{p_{G}\left(x\right)\cdot p_{EX}\left(x,z\right)+p_{data}\left(x\right)\cdot p_{GZ}\left(x,z\right)}$



**Proof of Theorem 1.** We can rewrite the mini-max problem of BiVi in Equation (Equation 3) as follows:
(3)$$\begin{array}{c} \min_{G,E}\max_{D}V\left(D,E,G\right)\\ =\min_{G,E}\max_{D}E_{z∼p_{z}}E_{\left(x,z\right)∼p_{EX}}\left[\log\left(D\left(x,E\left(x,z\right)\right)\right)\right]\\ \;+E_{x∼p_{data}}E_{\left(x,z\right)∼p_{GZ}}\left[\log\left(1-D\left(G\left(x,z\right),z\right)\right)\right]\\ \;+E_{x∼p_{data}}E_{\left(x,z\right)∼p_{GZ}}\left[L_{1}\left(CWT\left(x\right),CWT\left(G\left(x,z\right)\right)\right)\right] \end{array}$$
For fixed *G* and *E*, we denote the $\arg\max_{D}V$ that maximize *V* as C(G,E). In this case, C(G,E) can be expressed as an integral form in Equation (Equation 4) as follows:
(4)$$\begin{array}{cc} C(G,E)= & \int_{z}p_{z}\left(z\right)\int_{\left(x,z\right)}p_{EX}\left(x,z\right)\left[\log(D(x,z))\right]d\left(x,z\right)dz\\ \; & +\int_{x}p_{data}\left(x\right)\int_{\left(x,z\right)}p_{GZ}\left(x,z\right)\\ \; & \times \left[\log\left(1-D\left(x,z\right)\right)+L_{1}\left(x,z\right)\right]d\left(x,z\right)dx \end{array}$$
Because distribution of $p_{z}\left(z\right)$ is equal to distribution of $p_{G}\left(x\right)$, C(G,E), can be simplified in Equation (Equation 5) as follows:
(5)$$\begin{array}{cc} C(G,E)= & \int_{x}[p_{G}\left(x\right)\int_{\left(x,z\right)}p_{EX}\left(x,z\right)\left[\log(D(x,z))\right]\\ \; & +p_{data}\left(x\right)\int_{\left(x,z\right)}p_{GZ}\left(x,z\right)[\log\left(1-D\left(x,z\right)\right)\\ \; & +L_{1}\left(x,z\right)]]d\left(x,z\right)dx \end{array}$$
To obtain optimal *D*, we can rewrite the equation for *G* and *E* as equation for *D* in Equation (Equation 6) as follows:
(6)$$\begin{array}{cc} C(G,E) & =C^{\prime }\left(D\right)\\ \; & =\int \int a\cdot b\log(D)+c\cdot d\log(1-D)\\ \; & \;\;\;+L_{1}d\mu d\phi \end{array}$$
where, $a=p_{G}\left(x\right)$, $b=p_{EX}\left(x,z\right)$, $c=p_{data}\left(x\right)$, and $d=p_{GZ}\left(x,z\right)$. For the integral to be maximized, the integrand must be maximal. Let $f\left(D\right)=a\cdot b\log\left(D\right)+c\cdot d\log\left(1-D\right)+L_{1}$; then, via differentiation, $f^{\prime }\left(D\right)=a\cdot b/D+c\cdot d/\left(1-D\right)$. Because *f* has maximum value at $f^{\prime }=0$, we can obtain the following result:
(7)$$\begin{array}{cc} D^{*} & =\frac{a\cdot b}{a\cdot b+c\cdot d}\\ \; & =\frac{p_{G}\left(x\right)\cdot p_{EX}\left(x,z\right)}{p_{G}\left(x\right)\cdot p_{EX}\left(x,z\right)+p_{data}\left(x\right)\cdot p_{GZ}\left(x,z\right)} \end{array}$$
Therefore, *D* has optimal value of 1/2 when $p_{G}\left(x\right)\times p_{EX}\left(x,z\right)=p_{data}\left(x\right)\cdot p_{GZ}\left(x,z\right)$. □

#### 3.2.3. Convergence of C(G,E)

**Theorem 2.** 
*C(G,E) has global minimum of $P_{G}=P_{data}$*


**Proof of Theorem 2.** When rearranging the expression for C(G,E) at the optimal *D*, it is as expressed in Equation (Equation 8).
(8)$$\begin{array}{cc} C(G,E) & =\int_{x}[p_{G}\left(x\right)\int_{\left(x,z\right)}p_{EX}\left(x,z\right)\left[\log(D(x,z))\right]\\ \; & \;+p_{data}\left(x\right)\int_{\left(x,z\right)}p_{GZ}\left(x,z\right)\left[\log(1-D(x,z))\right]\\ \; & \;+L_{1}\left(x,z\right)]d\left(x,z\right)dx\\ \; & =\int_{x}\int_{\left(x,z\right)}p_{G}\left(x\right)p_{EX}\left(x,z\right)\left[\log(D^{*})\right]\\ \; & \;+p_{data}\left(x\right)p_{GZ}\left(x,z\right)\left[\log(1-D^{*})\right]\\ \; & \;+L_{1}\left(x,z\right)]d\left(x,z\right)dx \end{array}$$
When isolating the $L_{1}$ portion, it can be expressed in Equation (Equation 9) as follows:
(9)$$\begin{array}{c} L_{1}\left(x,z\right)=L_{1}\left(CWT\left(x\right),CWT\left(\hat{x}\right)\right) \end{array}$$The $L_{1}$ fundamentally approximates the output values by inputting both the original data and the generated data into the same function. Because the wavelet transformation is based on convolution operations via FFT, its inverse transformation can be considered. This indicates that the $L_{1}$ corresponding to the wavelet transformation operates mathematically in the same manner as the actual pre-transformation loss, approximating the generated data to the real data. Consequently, approximation through the $L_{1}$ has the effect that $P_{G}\approx P_{data}$. Via the converge of $L_{1}$, we can assume that $p_{G}\left(x\right)\approx p_{data}\left(x\right)=p\left(x\right)$; then, C(G,E) can be rewritten as follows in Equation (Equation 10).
(10)$$\begin{array}{cc} C(G,E) & =\int_{x}\int_{\left(x,z\right)}p_{G}\left(x\right)p_{EX}\left(x,z\right)\left[\log(D^{*})\right]\\ \; & \;+p_{data}\left(x\right)p_{GZ}\left(x,z\right)\left[\log(1-D^{*})\right]\\ \; & \;+L_{1}\left(x,z\right)d\left(x,z\right)dx\\ \; & =\int_{x}p\left(x\right)[\int_{\left(x,z\right)}p_{EX}\left(x,z\right)\log\left(D^{*}\right)\\ \; & \;+p_{GZ}\left(x,z\right)\log\left(1-D^{*}\right)]d\left(x,z\right)dx\\ \; & =\int_{x}p\left(x\right)\cdot [2JS(P_{EX}\left|\right|P_{GZ})-\log4]dx \end{array}$$
where the $L_{1}$ converges to 0 and consequently vanishes. The convergence of C(G,E) for this case has been proven in the previously presented BiGAN paper [21]. The minimum value for this case is as follows:
(11)$$\begin{array}{c} \;2JS(P_{EX}||P_{GZ})-\log4\gg -\log4\\ ∴C\left(G,E\right)\gg -\log4\cdot \int_{x}p\left(x\right)dx=-\log4 \end{array}$$In conclusion, the adversarial loss and $L_{1}$ for C(G,E) converge simultaneously. Owing to the properties of the two converging limits, C(G,E) converge to −log4.Therefore, the optimal *D* is $\frac{1}{2}$, and the generated data and the real data are approximated to have the same distribution as indicated by the above equation. □

## 4. Experiment

### 4.1. Experiment Setting

#### 4.1.1. Evaluation Method

When two probability distributions, P and Q, exist, KL divergence and JS divergence are representative quantitative metrics to measure the similarity of the two distributions. The KL divergence, or Kullback–Leibler divergence, calculates the difference between a reference probability distribution and a target probability distribution and is defined as follows [46].
(12)$$\begin{array}{c} KL\left(p∥q\right)=\left\{\begin{array}{c} \sum_{i}p_{i}\log\frac{p_{i}}{q_{i}}=-\sum_{i}p_{i}\log\frac{q_{i}}{p_{i}}\\ \int p\left(x\right)\log\frac{p(x)}{q(x)}dx=-\int p\left(x\right)\log\frac{q(x)}{p(x)}dx \end{array}\right. \end{array}$$
where *P* is the reference distribution and *Q* is the target distribution. The KL divergence has two notable characteristics, and one of them indicates that KL divergence cannot be used as a distance measure between two probability distributions.
(13)$$\begin{array}{c} KL(p∥q)\geq 0\\ KL(p∥q)\neq KL(q∥p) \end{array}$$

The inability to use KL divergence as a distance concept is a critical issue when employing it as an evaluation metric for AI models. This is because different values can emerge based on how the reference probability distribution is set. That is, the model performance can be measured differently based on whether the reference probability distribution is from the distribution of values generated by the model or the actual data distribution. Therefore, using KL divergence as a metric to measure the performance of an AI model can be challenging. Consequently, we utilize the Jensen–-Shannon divergence, which expands upon the concept of distance inherent in KL divergence [47].
(14)$$JS\left(p∥q\right)=\frac{1}{2}KL\left(p∥\frac{1}{2}\left(p+q\right)\right)+\frac{1}{2}KL\left(q∥\frac{1}{2}\left(p+q\right)\right)$$

JS divergence is defined as mentioned above, and because it satisfies the commutative property, it is suitable for representing the distance between two distributions. That is, in contrast with KL divergence, it outputs the same value regardless of which distribution is set as the reference point. Therefore, in this study, JS divergence was used as a quantitative evaluation metric for the proposed model.

#### 4.1.2. Dataset

This study aimed to generate 2D vibration data originating from rotating machinery. To evaluate the performance of the proposed BiVi-GAN, we conducted training and testing using the ‘Rotating Machinery Fault Type AI Dataset’ (Ministry of SMEs and Startups, Korea AI Manufacturing Platform (KAMP), KAIST, 23 December 2022). The dataset, as shown in Figure 8, was acquired at a rotational speed of 1500 RPM using a rotor testbed. The specifications of the equipment used for data collection are listed in Table 1.

The collected data consisted of four conditions: normal, imbalance, mechanical looseness, and a combination of imbalance and mechanical looseness. Imbalance refers to the most common cause of vibration in rotating machinery. It signifies a condition where the center of mass of the motor does not align with the center of its rotational axis. Mechanical looseness refers to a condition where the equipment is not securely anchored to a flat surface or ground, causing the misalignment or tilting of the equipment. In this study, the four conditions were organized into separate classes, and one-hot encoding was applied. This encoded information was then concatenated with the latent z, forming a class-based GAN structure.

The original data were collected over 140 s from four sensors with a phase difference of 90°. It consisted of a total of 3,772,385 points from four channels. For training, a pair of two sensors with a phase difference of 90° was configured. The window size was set to 1024, and slicing was performed using a stride of 200. In addition to slicing, no other signal processing or preprocessing techniques were applied. The data were used in their original state with considerable noise as input to maintain them in as similar as possible a state to the data collected in a real environment. The training dataset was composed of 7080 slices, with each class being represented equally. Additionally, a test dataset was formed with 2280 slices.

#### 4.1.3. Implementation

BiVi-GAN was implemented using the PyTorch framework, and experiments were conducted in the following computing environment. The generator, discriminator, and encoder of BiVi-GAN were constructed based on one-dimensional convolutional neural networks (1D-CNN) and implemented using a feature pyramid structure for both the encoder and discriminator. However, in the generator, the part responsible for generating cross-wavelet transform images was based on two-dimensional CNN (2D-CNN). It was structured with a network depth identical to the other networks. The generator, encoder, and discriminator networks used convolutional networks with a kernel size of four. Only the final layer of the discriminator included a sigmoid layer to make a binary discrimination between real and fake samples.

Training was conducted in batches, with each training iteration using a batch size of 128. The generator performed the CWT, and, owing to the inherent nature of GANs, where the generator tended to learn more slowly than the discriminator, there might be a risk of model collapse owing to differences in learning speeds. To compare and analyze the performance of the proposed BiVi-GAN, an ablation study was conducted. A total of six experiments were set up, each consisting of 500 epochs of training and validation under the same conditions. These experiments varied based on whether the CWT loss was used, whether the order frequency was used, and how many order features were chosen to utilize.

### 4.2. Ablation Study

The proposed BiVi-GAN is built on the foundation of the existing Bidirectional GAN and introduces two key contributions to advance the model. One of the contributions is the utilization of the PI input, which extracts the first-order phase information through order analysis from the input x and incorporates it into the model. The other contribution involves training the model’s ability to reconstruct the cross-wavelet transform and use it as an auxiliary task with a PI loss. An ablation study was conducted to assess the impact of these two methodologies on the model’s performance and determine the improvement compared with the baseline model. The analysis of the performance variation concerning the utilization of rotational synchrony components involves three conditions: using only the 1x component, using both the 1x and 2x components, and not using any component, thus examining the effect of these configurations on the PI input’s performance. Additionally, by examining the conditions of using and not using the cross-wavelet transform loss, the performance changes related to PI loss were analyzed. A total of six experiments were conducted, and the results of each experiment can be quantitatively analyzed using JS divergence. The JS divergence value approaching zero indicated that the data generated by the model were similar to the original data, while a value close to one indicated dissimilarity with the original data. The results of each experiment are listed in Table 2.

Based on the experimental results, the average JS divergence for the model using the PI input was 0.35, while the model that did not use it had an average JS divergence of 0.79. On average, this represents a 55.26% improvement in performance. Particularly, it should be noted that there was a 47.71% improvement in performance when using only the 1x component and 62.82% improvement when using both the 1x and 2x components. This suggests that using the PI input assisted in effectively capturing the vibration characteristics, contributing to improved performance. Furthermore, when using PI loss with cross-wavelet transform images, the average JS divergence was 0.45, which is approximately a 17.73% improvement compared with the model’s average of 0.55 when not using PI loss. Both sets of results provided experimental evidence that the two contributions proposed in BiVi-GAN indeed contributed to improved performance. Finally, when both PI input and PI loss are applied in BiVi-GAN, it achieves a JS divergence of 0.23. This represents a 70% improvement compared with the baseline model, Bidirectional GAN, which had a JS divergence of 0.77. It is evident that applying domain-specific expertise to data-driven deep learning models yielded significantly better performance compared with only using data.

### 4.3. Validation of GAN Data via Training

To verify whether the vibration data generated through the proposed BiV-GAN are effective as input data for actual learning-based methodologies, experiments were conducted to train one of the most representative deep learning models used in data analysis methodologies based on deep learning. The deep learning model used in the experiment utilizes a CNN model consisting of 3 Conv layers and 1 FC layer. The data were trained in the same learning environment, using the same model and the same hyperparameters as outlined in the following Table 3. The performance of the proposed methodology was validated using the learning dataset by training under three different dataset environments and performing tests on real data to evaluate the performance of the trained model.

In the first training, the learning was conducted using only the data actually acquired. In the second training, learning was performed using only the data generated through the proposed methodology, and in the final experiment, both the generated data and the actual data were utilized for the training. A classification task was carried out for a total of four classes (normal, imbalance, mechanical looseness, and a combination of imbalance and mechanical looseness), and to evaluate the quantitative performance, Precision, Recall, and F1 score were used. Precision represents the ratio of actual positives among the positive judgments made by the model, while Recall represents the ratio of positives correctly identified by the model out of the actual positives. The F1 score is obtained through the harmonic mean of Precision and Recall, with all three indicators having values between 0 and 1, where values closer to 1 indicate better model performance. The results of the experiments are listed in Table 4.

According to the experimental results, training with only real data resulted in an F1 score of 0.722, while training with only generated data achieved an F1 score of 0.683. When both types of data were used for training, an F1 score of 0.816 was achieved, resulting in an approximate 13-percent improvement in performance, marking the highest performance achieved. These results demonstrate that in cases where there is insufficient or a lack of diversity in the available data, using the proposed BiVi model to synthesize data and incorporating it into the training can be beneficial, especially for utilizing bearing vibration data.

## 5. Conclusions

In this study, we proposed BiVi-GAN for generating vibration data in rotating machinery. Our proposed model leveraged the strengths of deep learning in capturing data characteristics and combined them with the physical knowledge of PINN. Consequently, we were able to generate data that closely resembled real-world data. Using the cross-wavelet transform as PI loss and order-analysis results as PI input, we observed a dramatic improvement in performance, particularly when applying PI input. In the future, we aim to conduct a study on methodologies that allow the application of physical information to other tasks. We intend to explore how expertise in machine engineering, where substantial physical knowledge is required, can be effectively complemented by AI technologies.

## Figures and Tables

**Figure 1 sensors-24-01765-f001:**
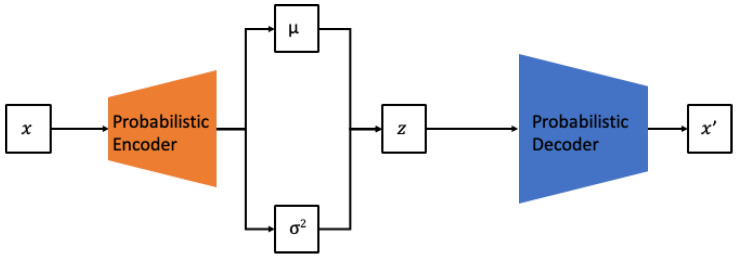
Architecture of VAE.

**Figure 2 sensors-24-01765-f002:**
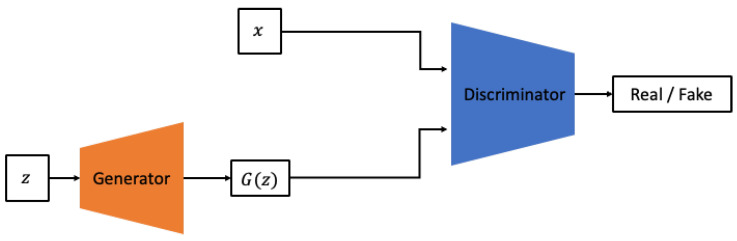
Architecture of GAN.

**Figure 3 sensors-24-01765-f003:**
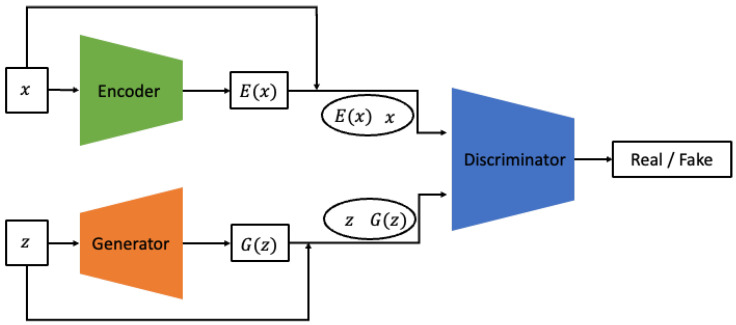
Architecture of Bidirectional GAN.

**Figure 4 sensors-24-01765-f004:**
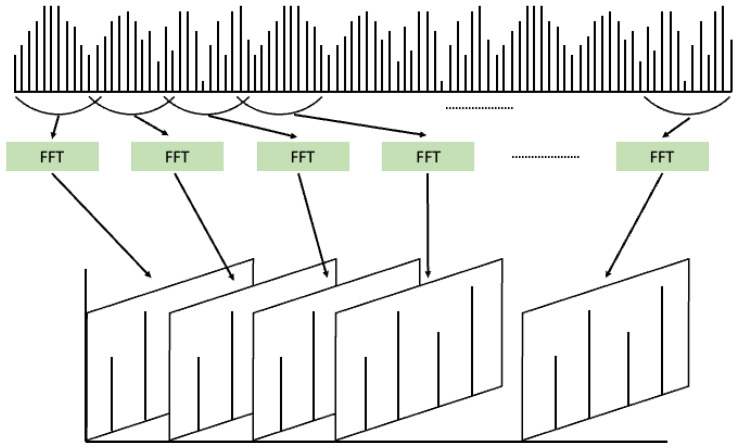
Short-Time Fourier Transform (STFT).

**Figure 5 sensors-24-01765-f005:**
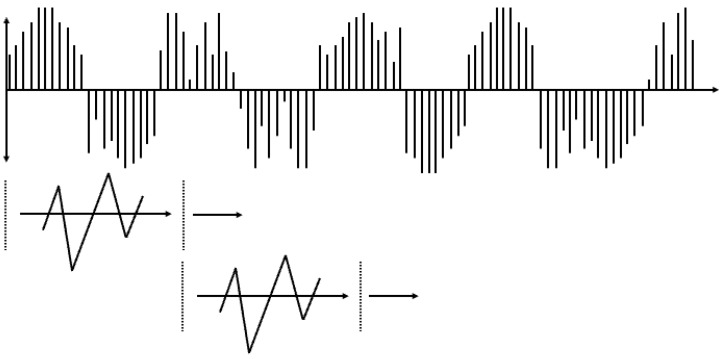
Wavelet Transform.

**Figure 6 sensors-24-01765-f006:**
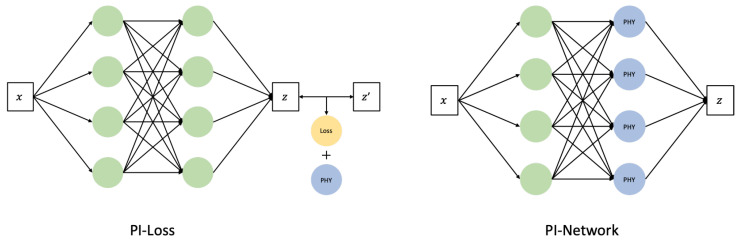
(**Left**): Physics-informed loss model; (**Right**): Physics-informed architecture.

**Figure 7 sensors-24-01765-f007:**
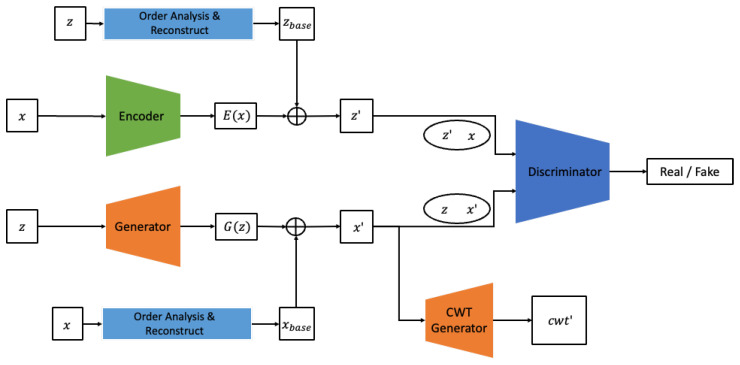
Architecture of BiVi-GAN.

**Figure 8 sensors-24-01765-f008:**
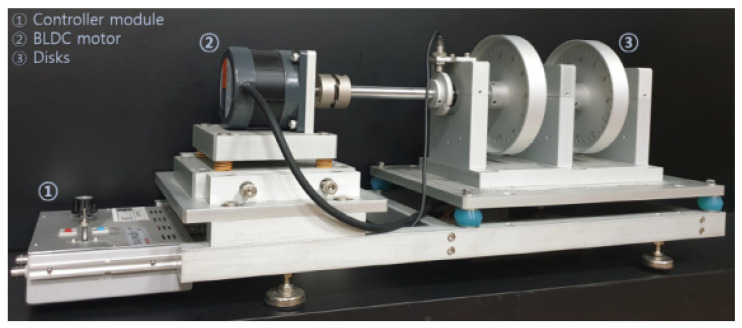
Rotor testbed.

**Table 1 sensors-24-01765-t001:** Rotor testbed setting.

Size	673 mm (W) × 280 mm (D) × 281 mm (H)
Weight	25 kg
Material	Aluminum
Bearing	6202ZZ × 2EA
Motor	DC 12V × 0.25HP (0.2 W), 0~3000 RPM
Main power	220 VAC

**Table 2 sensors-24-01765-t002:** Ablation study result.

Experiment No.	Order Analysis	Cross-Wavelet Transform	JS Div
1	x	o	0.774
2	x	x	0.821
3	1x	o	0.360
4	1x	x	0.474
5	2x	o	0.230
6	2x	x	0.363

**Table 3 sensors-24-01765-t003:** Train parameter setting.

Parameter	Setting
o	0.774
x	0.821
o	0.360
x	0.474
o	0.230
x	0.363

**Table 4 sensors-24-01765-t004:** Data validation experiment result.

Experiment No.	Train Data	Precision	Recall	F1 Score
1	Real Only	0.728	0.715	0.722
2	Generated Only	0.681	0.685	0.683
3	Real + Generated	0.815	0.816	0.816

## Data Availability

The data presented in this study are available in KAMP (Korea AI Manufacturing Platform) at https://www.kamp-ai.kr/aidataDetail?.

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
