# Peer review of "BiVi-GAN: Bivariate Vibration GAN"

_sensors, 2024, doi:10.3390/s24061765_

Round 1

Reviewer 1 Report

Comments and Suggestions for Authors

The paper presents a model of a Bivariate Vibration Generative Adversarial Networks (BiVi-GAN).

The authors say that ”…the BiVi-GAN showed a 70% performance improvement in terms of JS divergence compared to the baseline biwavelet-GAN model.

The paper is well structured, well developed, presenting good references.

The sections three and four present the methodology and testing, showing good results, what permits the authors to conclude that “using cross wavelet transform as PI-loss and order-analysis results as PI-input” the authors observed a dramatic improvement in performance, particularly when applying PI-input.

Having into account the preceding I consider the paper can be published with any changes.

Author Response

Dear Reviewer,

We would like to express our sincere gratitude for your constructive comments and positive feedback on our manuscript, "Model of Bivariate Vibration Generative Adversarial Networks (BiVi-GAN)." We are delighted to hear that you found the paper well-structured and the methodology and testing sections well-developed.

We are honored by your assessment that the paper can be published without any changes. However, We added 2.4 Bearing Fault Detection via Deep Learning Model (line 197) under the Related Work part to explain the fact that the proposed methodology is related to bearing failure diagnosis. Additionally, an additional experiment was conducted to prove the usefulness of the data generated by the proposed BiVi-GAN in model learning, and details about this can be found in Experiment section 4.3 Validation of GAN Data via Training (line 437) .

Thank you once again for your valuable input and for considering our work suitable for publication. We are looking forward to the opportunity to contribute to the academic community through your esteemed journal.

Sincerely

Reviewer 2 Report

Comments and Suggestions for Authors

The article presents an approach to augment vibration data for rotating machinery using a generative model called BiVi-GAN. The article claims that BiVi-GAN outperforms the baseline model in terms of JS divergence and shows the potential of combining physics-informed neural networks with data-driven AI models for prognosis and health management (PHM). Nonetheless, the results are not sufficient enough to prove the claims. There is a section (line 307) before the conclusion that discusses about the results of the experiment, however this should be expanded and the tests could have been more detailed. The paper lacks the detailed results section and that paragraph would likely counts as a discussion rather than a detailed results analysis.

I would recommend the author to expand upon their reults

Author Response

Dear Reviewer,

Thank you for taking the time to review our article and providing valuable feedback. We appreciate your insights and recognize the importance of your suggestions regarding the presentation of our experimental results.

We understand your concern that the results presented do not sufficiently support the claims made about the effectiveness of the BiVi-GAN model. In response to your feedback, we have expanded the section before the conclusion, we conducted additional experiment to prove the usefulness of the data generated by the proposed BiVi-GAN in model learning, and details about this can be found in Experiment section 4.3 Validation of GAN Data via Training (line 437) .

Furthermore, We added 2.4 Bearing Fault Detection via Deep Learning Model (line 197) under the Related Work part to explain the fact that the proposed methodology is related to bearing failure diagnosis.

We believe these revisions and expansions address your concerns and improve the paper's clarity and depth, particularly in the results section. We hope that these changes meet your expectations and strengthen the manuscript.

Thank you once again for your constructive criticism and guidance. We look forward to your further feedback.

Sincerely

Reviewer 3 Report

Comments and Suggestions for Authors

Comments on the Quality of English Language

Minor editing of English language required

Author Response

Dear Reviewer,

Thank you very much for your detailed review and constructive suggestions on our manuscript regarding the bivariate vibration generative adversarial networks (BiVi-GAN) model. We are grateful for your positive comments about the structure and results of our paper.

We have taken your feedback seriously and have made the following revisions to enhance the quality of our manuscript:

  1. We added 2.4 Bearing Fault Detection via Deep Learning Model (line 197) under the Related Work part. This includes a detailed discussion on the vibration signal characteristics specific to bearing faults. We have also referred to the suggested literature "A Robust Deep Learning Network for Low-speed Machinery Fault Diagnosis based on Multi-kernel and RPCA" to enrich our discussion and ensure our research is well-grounded in the latest findings.
  2. In response to your second suggestion, we have revised the literature review section to better highlight the research gaps that our paper aims to address. We have also included a more detailed discussion on the main contributions and novelty of our paper in 2.4 Bearing Fault Detection via Deep Learning Model (line 197). Additionally, we have incorporated discussions on the emerging areas of deep learning and multiple sensors in fault diagnosis across various industrial applications, referencing the suggested works including "A Novel Convolutional Neural Network for Low-speed Structural Fault Diagnosis under Different Operating Conditions" and "Feature Extraction of Multi-sensors for Early Bearing Fault Diagnosis Using Deep Learning Based on Minimum Unscented Kalman Filter."
  3. We have improved the resolutions of all figures within the manuscript.
  4. We have thoroughly reviewed the entire manuscript for grammatical errors again.

We hope that these revisions address your concerns and significantly improve the manuscript. We appreciate your suggestions as they have undoubtedly enhanced the quality and clarity of our work.

Thank you once again for your invaluable feedback and guidance.

Sincerely

Round 2

Reviewer 2 Report

Comments and Suggestions for Authors

The authors modified and added all the recommended changes to the manuscript.